# From COPD to Smoke-Related Arteriopathy: The Mechanical and Immune–Inflammatory Landscape Underlying Lung Cancer Distant Spreading—A Narrative Review

**DOI:** 10.3390/cells14161225

**Published:** 2025-08-08

**Authors:** Giulia M. Stella, Francesco Rocco Bertuccio, Cristina Novy, Chandra Bortolotto, Ilaria Salzillo, Fabio Perrotta, Vito D’Agnano, Valentina Conio, Vittorio Arici, Pietro Cerveri, Andrea Bianco, Angelo Guido Corsico, Antonio Bozzani

**Affiliations:** 1Department of Internal Medicine and Medical Therapeutics, University of Pavia Medical School, 27100 Pavia, Italy; francesco.bertuccio01@gmail.com (F.R.B.); cristina.novy01@universitadipavia.it (C.N.); angelo.corsico@unipv.it (A.G.C.); 2Unit of Respiratory Disease, Cardiothoracic and Vascular Department, IRCCS Policlinico San Matteo, 27100 Pavia, Italy; v.conio@smatteo.pv.it; 3Diagnostic Imaging and Radiotherapy Unit, IRCCS Policlinico San Matteo, 27100 Pavia, Italy; chandra.bortolotto@unipv.it; 4Department of Clinical, Surgical, Diagnostic, and Pediatric Sciences, University of Pavia Medical School, 27100 Pavia, Italy; 5Unit of Vascular Surgery, Cardiothoracic and Vascular Department, IRCCS Policlinico San Matteo, 27100 Pavia, Italy; ilaria.salzillo01@universitadipavia.it (I.S.); v.arici@smatteo.pv.it (V.A.); a.bozzani@smatteo.pv.it (A.B.); 6Department of Translational Medical Sciences, University of Campania L. Vanvitelli, 80131 Naples, Italy; fabio.perrotta@unicampania.it (F.P.); vito.dagnano@unicampania.it (V.D.); andrea.bianco@unicampania.it (A.B.); 7U.O.C. Clinica Pneumologica L. Vanvitelli, Monaldi Hospital, A.O. dei Colli, 80131 Naples, Italy; 8Department of Industrial and Information Engineering, University of Pavia, 27100 Pavia, Italy; pietro.cerveri@unipv.it

**Keywords:** lung cancer, COPD, arteriopathy, inflammation, atherosclerosis, immune checkpoint inhibitors, triple therapy

## Abstract

Metastatic dissemination defines a complex phenomenon driven by genetic forces and, importantly, determined by interaction between cancer cells and the surrounding stroma. Although the biologic and immune reactions which characterize the process have been widely and extensively evaluated, fewer data are available regarding the mechanical and physical forces to which circulating neoplastic clones are exposed. It should be hypothesized that this interaction can be modified in case of concomitant pathologic conditions, such as chronic vasculopathy, which frequently occurs in lung cancer patients. We here aim at analyzing and discussing the complex interplay between lung malignant transformation and arteriopathy, mainly focusing on the immune–inflammatory systemic reaction. Notably—in most instances—smoking-related fixed airflow obstruction, including but not limited to COPD, frequently coexists and contributes to both tumor progression and vascular complications. Attention is paid mainly to the analysis of the role of immune checkpoint inhibitors and their interaction with triple bronchodilation and antiaggregants. Understanding the biomechanical and molecular dynamics of lung cancer progression in altered vascular territories has several translational implications in defining risk stratification and in surgical planning and therapeutic targeting. Moreover, computational modeling of the physical forces which regulate the transit and extravasation of metastatic clones in altered contexts could be of help in deciphering the whole process and in determining more effective blockade strategies.

## 1. Introduction

Although scientific knowledge of the onset and progression of lung cancer (LC) has significantly increased in recent decades, with favorable implications for the outcome of affected patients, it is confirmed worldwide as one of the big killers, primarily due to its high metastatic potential and the frequent occurrence of advanced-stage diagnosis [1,2,3,4].

Unequivocal and consolidated epidemiological and experimental evidence confirms tobacco smoking as the most important risk factor for the development of lung cancer [5,6,7,8,9], although individual susceptibility [10,11,12,13,14,15,16] and other concomitant lung pathologies [17,18,19,20] impact LC incidence. There is a close correlation between the onset of the disease and the number of cigarettes/day smoked, and with the duration of the habit in years, the depth with which the smoke is inhaled, and the tar and nicotine content of the cigarettes smoked [21]. The relative risk of smokers of getting LC is 14 times higher than that of non-smokers, while for heavy smokers (for those who consume more than 25 cigarettes/day), it rises to 30 times. For those who quit smoking, the risk progressively reduces over the following 7–10 years, after which it returns to overlap that of non-smokers [22]. Exposure to passive smoking is recognized as risk factor, mainly if it occurs at a young age (under 25 years) [23]. It is more complicated to define the pathogenetic role of air pollution, given the complexity and heterogeneity of the toxicants that can be detected; however, in residents of urban areas, the risk of having LC is 1.2–2.3 times greater than in residents of rural areas [24,25,26,27,28].

Importantly, several large-scale population studies have demonstrated that fixed airflow obstruction—even in the absence of a formal COPD diagnosis—is an independent risk factor for lung cancer and cardiovascular morbidity. This relationship is mediated not only by shared exposures such as smoking, but also by overlapping biological pathways including chronic inflammation, tissue remodeling, and systemic vascular impairment. With respect to the clinical presentation and molecular features of smoke-associated LC, most data actually relate to non-small-cell LC (NSCLC), and in this review the term LC refers to NSCLC. The complex natural history of LC implies that distant metastases are most often detectable at the time of clinical diagnosis. Metastatic progression is driven by genetic programs based on the dynamic crosstalk between genes and the distant microenvironment [29,30,31]: it is generally a late event but, in some cases, it starts early and prevails over the growth of the primary lesion [32]. Overall, metastatic clones need to acquire the biological properties required to detach from the mass, invade blood (or lymphatic) vessels, survive in the flow in the absence of anchorage, and then extravasate and colonize distant organs. Moreover, it is now well known that the immune system interacts closely with tumors throughout the entire process of malignant transformation, and the last decade has seen the rapid development of immunotherapy and its role as a crucial strategy in the treatment of cancer, especially LC. A smoking habit acts as risk factor for chronic vascular disease as well. Thus, LC patients are frequently co-affected by chronic obstructive pulmonary disease (COPD) and arteriopathy. Chronic obstructive pulmonary disease (COPD), a common consequence of long-term tobacco exposure, is associated with profound changes in vascular structure and function, systemic inflammation, and immune dysregulation [33,34,35]. Smoke-related vasculopathy may profoundly alter the mechanical and immunologic microenvironment [36], hypothetically modifying the success of metastatic colonization at distant sites [37,38]. Although both conditions are well characterized and treated, few data are available regarding the complex interaction between them and LC in terms of mechanistic and therapeutic reciprocal interferences and effects. Thus, the aim of this work is to point out from data in the available literature the interplay between the two chronic conditions and LC progression, which shares with them smoke as a risk factor with significant therapeutic implications. This knowledge gap becomes even more relevant in the context of chronic comorbidities frequently observed in lung cancer patients, especially those with a history of heavy smoking. Moreover, a deeper understanding of the fate of LC metastatic cells in the blood flow in a real-life context should be of help in designing and modulating innovative systemic diagnostic tools, surgical strategies, and drug delivery technologies.

## 2. Methods

We performed an extensive search using the following biomedical databases: Medline using the PubMed interface, Web of Science, and Embase. Search terms were as follows: “lung cancer” and “chronic pulmonary obstructive disease—COPD” and “vascular disease”, “aneurysm”, “arteriopathy”, “metastases”, “immunotherapy”, “triple bronchodilation”; “antiaggregants” and “circulating tumor cells” and “metastatic niche”. No restriction on publication date was applied, and the last search was performed on 11 July 2025. Only articles written in or translated into English were included. The search for each topic was performed independently by pneumo-oncologists, pulmonologists, radiologists, and vascular surgeons focusing on their areas of interest.

## 3. The Lung Region

### 3.1. The Complex Landscape Linking COPD to Malignant Transformation and Cardiovascular Disease

#### 3.1.1. Inflammation and Tissue Remodeling

The intricate interplay between chronic smoking, COPD, and systemic vasculopathy forms a complex and pro-inflammatory landscape that should be hypothesized to influence both carcinogenesis and cancer progression in the lung. Cigarette smoke represents a multifaceted insult, rich in oxidants, reactive nitrogen species, and carcinogens, which chronically activates epithelial cells, fibroblasts, macrophages, and endothelial structures. The result is a persistent inflammatory milieu that extends well beyond the airway architecture, involving the pulmonary vasculature and systemic endothelium [36]. Matrix metalloproteinases (particularly MMP-2 and MMP-9), commonly elevated in COPD, degrade the extracellular matrix and basal membranes, removing critical barriers to tumor invasion. This ECM breakdown, driven by both inflammatory and hypoxic stimuli, contributes to a permissive metastatic niche within the lung and possibly at distant vascular sites in smokers. COPD is a heterogeneous disease characterized by two key features. The first is innate immune activation with macrophages and some polymorphs in the airway lumen, in particular (more than in the subepithelial airway wall), and there might be some activation of the adaptive system, especially late in the process [39]. Secondly, it is defined by the important role of microbial colonization/infection [40]. It should be underlined that critical cellular orchestrators of inflammation are cells of the innate immune system, such as neutrophils and macrophages [41]. In other words, cells and tissues—as epithelial and endothelial layers—can respond to a variety of stimuli and can interact mutually with inflammation, but that is not inflammation in itself. Lung cancer and COPD share a number of biological processes, such as the above-described ones and especially that of epithelial mesenchymal transition (EMT) [42,43]. In some instances, inflammatory reaction is also related to airway remodeling or EMT, a major and central event starting from the early COPD phases [44,45,46]. EMT is a dynamic and reversible biological process in which epithelial cells lose their apicobasal polarity and cell–cell adhesion properties, acquiring mesenchymal characteristics such as enhanced motile properties and resistance to apoptosis. EMT is a critical cellular program in which epithelial cells lose their polarity and adhesion, acquire mesenchymal traits, activate epithelial neovascularization, and gain enhanced migratory and invasive capabilities. Thus, EMT can occur in physiological conditions, as during embryo development, namely, type I EMT [47], or during wound healing, tissue regeneration, and organ fibrosis (type II EMT) [48]. Aberrant activation of EMT (type III EMT) orchestrates cancer progression in terms of promotion of clonal evolution but also in the generation of a pro-metastatic stroma [49,50,51]. Smoke is implicated in EMT activation in both lung cancer and COPD [52]. In this context, inhaled corticosteroids (ICSs), mainly at higher doses, should mitigate LC risk reduction, although in vivo studies are needed [53,54].

In COPD, EMT is observed early in disease progression, particularly in basal stem/progenitor cells, and contributes to airway wall thickening, subepithelial matrix deposition, and fibroblast activation. Chronic exposure to smoke, repeated microbial colonization, and hypoxia all promote EMT through TGF-β, IL-6, and Wnt signaling pathways. EMT also facilitates immune escape and supports the generation of cancer stem-like cells, contributing to lung carcinogenesis and metastasis. EMT plays a pivotal role in embryogenesis and tissue repair, and, in COPD, it is associated with subsequent myofibroblast proliferation and airway fibrosis, deposition of matrix proteins, and remodeling ultimately leading to thickening of all layers constituting the subepithelial structures; overall, these processes link COPD to cancer onset [55,56].

#### 3.1.2. Gene Reprogramming

Mechanisms that disrupt epithelial cell function in patients with COPD are complex and not completely elucidated. However, the oxidative stress which drives the disease interferes with the balance of self-renewal and differentiation of stem/progenitor cells, thus impairing a proper repair of lung injuries [57]. This mechanism is certainly vital in COPD epithelial and subepithelial pathology, and is very likely to be under epigenetic control. Smoking habit and COPD are associated with epigenetic modification, which modulates DNA expression in different cells in the lungs and in the blood and is implicated in disease progression [58,59]. Chronic smoke exposure induces a range of genetic alterations, including mutations in *KRAS*, *TP53*, and chromosomal instability. However, growing attention is being paid to the role of epigenetic reprogramming—particularly in basal airway progenitor cells—in shaping both COPD pathology and lung cancer risk. Basal cells, which act as stem/progenitor elements in the bronchial epithelium, are susceptible to gene expression reprogramming driven by DNA methylation (e.g., hypermethylation of tumor suppressor loci), histone modifications (e.g., H3K27 acetylation), and dysregulated microRNAs (e.g., miR-21 and miR-34a). These epigenetic changes result in persistent loss of epithelial identity, enhanced mesenchymal traits, and acquisition of proliferative and migratory capacity—hallmarks of EMT. The remodeled airway epithelium, particularly in proximal bronchi, thus becomes fertile ground for both fibrosis and malignant transformation. Importantly, some of these epigenetic changes persist even after smoking cessation, suggesting a lasting imprint on the progenitor cell compartment. Moreover, interactions between microbial colonization, inflammatory cytokines (e.g., IL-6 and TGF-β), and epigenetic regulators contribute to long-term alterations in chromatin structure and transcriptional programs, amplifying the risk of neoplastic progression. Several genes encoding for inflammatory-related molecules (e.g., NF-κB, P38 MAPK, and IFN-β pathways) have been reported to be overexpressed, together with those related to lung injury pathways (*IL1R1*, *PTGS2*, *C3*, and *TLR4*) [60,61], whereas others are known to be downregulated, as the hedgehog interacting protein (*HHIP*), located on chromosome 4q31 [62]. Notably, the loss of *CDH1*, the gene encoding for E-cadherin, has been causally associated with chronic lung disease [63,64]. Fibroblasts, as well, are functionally different in COPD, according to specific epigenetic signatures involving induced pluripotent stem cells (iPSCs) [65].

#### 3.1.3. COPD and Cardiovascular Events

The anatomical and functional interplay between the lungs and the cardiovascular system implies that any dysfunction in one system is reflected in the other. The main relationship of a set of relationships also regards COPD, or, indeed, fixed airflow obstruction short of a clinical diagnosis, and cardiovascular disease in terms physical constraints due to large pressure fluctuations on the heart and large vessels. Fixed airways obstruction contributes to chronic lung hyperinflation and dynamic changes in intrathoracic pressure, which may impose cyclical mechanical strain on the heart and great vessels, promoting left ventricular dysfunction and vascular remodeling. These large pressure fluctuations, compounded by systemic hypoxia and oxidative stress, accelerate endothelial damage and atherosclerotic progression. Furthermore, persistent airflow limitation is associated with systemic inflammation and increased prevalence of clonal hematopoiesis—two factors that contribute to both lung cancer development and cardiovascular events. Multiple mechanisms underlie this association, including chronic systemic inflammation, endothelial dysfunction, and mechanical stress from exaggerated intrathoracic pressure swings. In particular, repetitive cycles of alveolar collapse and airway compression during forced expiration impose cyclic mechanical loading on adjacent vascular structures. This contributes to arterial stiffening, local tissue hypoxia, and endothelial injury—factors that collectively facilitate neoplastic initiation and vascular comorbidities. These biomechanical–vascular interactions remain under-recognized but may explain the high incidence of cardiovascular disease in lung cancer patients with preserved spirometry and significant smoking history. It is well known that COPD patients feature steeper blood pressure changes than the healthy population, and that the speed of fluctuations is associated with the severity of airflow limitation. Increased systolic blood pressure sustains the association between COPD and cardiovascular disease [66]. Most importantly, it is well known that airflow limitation is significantly associated with higher risk of death from myocardial infarction, irrespective of other factors such as age, sex, and smoking history [67]. Even though smoke is a shared risk factor, a chronic obstructive condition behaves as an independent factor for mortality, and patients with mild COPD actually have a higher probability of dying of a cardiovascular cause than of respiratory failure [68]. COPD is also associated with heart failure and pulmonary hypertension and affects cardiac function at rest and in reducing exercise performance [69]. With respect to the vascular system, arterial stiffness is increased in COPD, mainly related to both systemic inflammation and impaired endothelial nitric oxide (NO) production [70]. These data are coherent with the above discussed higher risk of hypertension that is associated with this condition [71]. Moreover, the occurrence of arterial aneurysmal disease and rupture is most likely associated with the condition of bronchial chronic obstruction [72,73]. The latter also impacts the outcome after endovascular repair (EVAR) [74].

### 3.2. Lung Cancer in Smokers

#### 3.2.1. Smoke-Induced Tumorigenesis

Cigarette smoking has a direct carcinogenic effect (genotoxic damage) due to the various substances produced during combustion. Among the components of cigarette smoke, the main carcinogenic action is attributed to polycyclic aromatic hydrocarbons, indirect carcinogens that require transformation into active intermediates by microsomal enzymes present at the level of bronchial cells [75,76,77]. Cigarette devices and vaping fluids, as well, have been shown to contain a number of carcinogens classified as both definite and probable, including nicotine derivatives (e.g., nitrosonornicotine and nitrosamine ketone), polycyclic aromatic hydrocarbons, heavy metals (including organometallic compounds), and aldehydes/other complex organic compounds. These molecules are present both in the e-liquid (with many aldehydes and other complex organic compounds used as flavors) and as a result of pyrolysis/complex organic reactions in the e-cigarette device (including clear carcinogens such as formaldehyde, formed by the pyrolysis of glycerol). Various studies demonstrate the transforming and cytotoxic activity of these derivatives in vitro and in vivo. The use of e-cigarette devices is significantly increasing, particularly among younger people and previous non-smokers. Considering the latency times (extrapolated from data on tobacco smoking), which reach up to 20 years, it is highlighted how this type of exposure can have very significant future implications for public health [78,79,80,81]. Moreover, many substances demonstrate an additive and sometimes synergistic effect with tobacco smoke. This is true, for example, for exposure to asbestos fibers, since it has been shown that non-smoking workers in the asbestos industry display a risk of developing LC 5 times higher than non-exposed workers and non-smokers, while the risk of exposed workers and even current smokers rises to 95 times [82,83]. Lung cancer cells express the nicotinic acetylcholine receptors (nAChRs) [84], and through the α7-nAChRs, nicotine can increase the invasive potential of LC cells, whereas it also promotes the expression of genes involved in epithelial to mesenchymal transition (EMT) in several cancer types [85,86,87,88,89]. Overall, experimental evidence suggests that smoke inhibits vascular collagen by reducing local production of prolyl-4-hydroxylase [90] and impairs matrix homeostasis by altering production of metalloproteinase and increasing T-cell infiltrates in response to the smoke-induced injury [91,92]. Smoke seems also to promote hyperplasia of the tunica intima as well as senescence of vascular smooth cells [93]. Nicotine promotes aneurysm formation by activating AMP-activated protein kinase α2 (AMPK-α2) and miR-21, probably in response to cellular stress and hypoxia [94]. Smoke exposure is also associated with the induction of a number of indirect genotoxic effects [95,96,97,98,99,100]: (i) alteration of DNA repair mechanisms and the consequent failure to repair to direct genotoxic damage; (ii) expression of polymorphisms of genes coding for enzymes involved in the metabolization process with increased exposure to genotoxic and carcinogenic agents; (iii) induction of the expression of genes involved in the pathogenetic mechanisms of inflammatory processes, oxidative stress, and tissue repair, which ultimately leads to chronic tissue damage and activation of molecular mechanisms promoting neoplastic progression and inflammatory infiltrates [101].

#### 3.2.2. Molecular Basis of Smoke-Associated Lung Carcinogenesis

Although a deep analysis of the molecular profile of LC in smokers goes beyond the scope of this review, some issues deserve to be recalled. Chromosomal alterations (e.g., promoter hypermethylation and loss of heterozygosity) are more frequent in tumors of smokers, and among these more frequent in squamous cell cancers than in adenocarcinomas. The genetic asset of LC in smokers is mainly addicted to somatic mutations in the *KRAS* gene (mainly affecting exon 12), correlating with poor survival. *KRAS* mutational frequency is allele-/tissue-specific and associated with smoking habit, although it is not described in small-cell lung cancers (SCLCs). *KRAS* alleles are non-uniformly distributed across cancers; they have different mutagenic origins related to exposure to tobacco smoke. The *KRAS* alleles have distinct co-mutation networks with mutual exclusion with *EGFR* mutations [102,103]. Once defined as an undruggable genetic driver, more recently, the activated *G12C KRAS* has become actionable via a newly identified switch II pocket, and an innovative class of small molecules has been developed. Thus, sotorasib and adagrasib are now approved in locally advanced or metastatic *KRAS*_G12C_ NSCLC [104,105,106,107]. Moreover, *p53* mutations are more common in NSCLCs (SCCs) in smokers, independently of *KRAS* and *EGFR* status [108]. Smoking also affects epigenetic status, and more than 2600 cytosine–phosphate–guanine sites (CpGs) are statistically significantly differentially methylated in smokers [109]. Exposure to cigarette smoke on the human airway epithelial cell is known to affect transcriptome profile as well. A number of data records on hierarchical clustering analysis performed on samples derived from current- and never-smoker samples have documented that in smokers some genes are overexpressed, such as those involved in the regulation of drug metabolism and oxidation–reduction reactions, mucous secretion, and some oncogenes (RAS pathway); tumor growth and inflammation (cystatin, *IL-8*, and *CD55*); fibroblast activation and proliferation (*HBP17*); and cell-cycle progression and control (*TOB1*, *DUSP6*, and *BRD2*). Unexpressed genes are those involved in negative regulation of inflammatory processes (antioxidant defence gene *BACH2* and *COX5B*), tumor suppressor genes, ubiquitination (*UBE2D2*), and DNA repair (*GTF2H3*). In particular, aberrant transcripts (deletions, LOH, and allelic loss) of *FHIT* (fragile histidine triad) have been described in NSCLCs (SCCs), directly related to smoking exposure. *FHIT* acts as a tumor suppressor and is localized in the chromosomal region 3p14.2, close to the most common fragile site of the human genome *FRA3B*; loss of function of the FHIT protein is associated with increased proliferation and reduction in the apoptosis index and is an independent indicator of clinical outcome [110,111,112,113,114]. It should be also remarked that even upon cessation of exposure to smoking, in some instances gene expression is irreversibly altered, thus justifying the persistence upon cessation of the risk of lung disease such as cancer [115,116,117,118]. Solid tumors consist of multiple cell types that differ in their state of differentiation and contain a cellular subset with phenotypic stemness characteristics, namely cancer stem cells (CSCs) which are characterized by the capacity of self-renewal and differentiation and which constitutively express non-specific molecular markers of multidrug- and radio-resistance; they retain the exclusive ability to support tumorigenesis (for a review, see [119,120,121,122,123]). It is conceivable that anatomically distinct CSC niches exist in the lung: in the proximal tract, elements carrying stem-like potential can be identified in basal cells (expressing keratins K5/K14+) [124], whereas in the peripheral tract, the morphological identification is more difficult, based on the expression of TTF1—thyroid transcription factor 1 [125], and corresponds to club cells (CC10+) in bronchioles, alveolar type II pneumocytes (SP-C+), and club cell variants co-expressing CC10 and SP-C [126,127,128]. Chronic cigarette smoking results in lung inflammation and epithelial damage that ultimately activate a chronic wound repair program. It induces, as widely reported, in the upper airways [129,130] the formation of multifocal epithelial lesions carrying genetic alterations such as LOH in 17p, 3p, 9p, 8p, 18q, and 11q13 amplification. Thus, it is likely that in smokers the phenomenon of “field cancerization” [131,132,133] culminates in malignant transformation, starting from precursor stem cells which define pre-neoplastic cyto-histologic settings. This model is coherent with the risk of lung cancer recurrence after radical surgery and the arousal of second primary or synchronous tumors, and it justifies the rationale for monitoring pre-neoplastic “field” lesions [134,135]. Moreover, recent studies have demonstrated the ability of bone marrow-derived stem cells to respond to epithelial wounding and contribute to epithelial repair through directly crossing lineage or fusion with lung epithelial cells and sustaining malignant transformation induced by the carcinogens in cigarette smokes [136,137]. Cigarette smoke contributes not only to malignant transformation but also to tumor progression by promoting (seed) and creating a favorable substrate (soil) to the process of neoplastic spreading and metastasis growth (Table 1) [138,139,140,141,142,143,144,145,146,147,148,149,150]. Overall, the strong immunogenicity associated with cigarette smoking is a reliable predictor of response to immune checkpoint inhibitors in LC. The increased number of mutations associated with the cellular damage exerted by smoke toxins is associated with the increased release of neoantigens, which may improve response to immunotherapy. From this perspective, smoke and smoke-induced mutational load are a proxy for tumor response [151]. Furthermore, hypoxia (a common feature in advanced COPD) plays a dual role in modulating tumor biology. Hypoxic stress in the lung activates hypoxia-inducible factors (HIFs) that promote angiogenesis, epithelial–mesenchymal transition (EMT), and immune evasion in neoplastic clones. Simultaneously, hypoxia exacerbates endothelial injury and promotes local and systemic vasoconstriction, further impairing vascular homeostasis. The end result is a “double-hit” phenomenon, where the vasculature is simultaneously primed for neoplastic invasion and incapable of mounting an effective barrier [38]. Another critical dimension is the profound impact of smoking and COPD on the immune–inflammatory landscape. Both are associated with systemic neutrophilic activation, elevated levels of TNF-α, IL-6, and IL-1β, and reduced regulatory T-cell activity. These alterations skew immune responses toward a pro-tumorigenic, tolerogenic profile. Moreover, smoking is a major inducer of clonal hematopoiesis and immunosenescence, which could further impair antitumor immunity and impact the efficacy of immune checkpoint inhibitors (ICIs) [33].

## 4. Vascular Compartment

### 4.1. Smoke-Induced Vascular Pathology

COPD generates a sustained pro-inflammatory environment with increased levels of TNF-α, IL-6, VEGF, and reactive oxygen species. These mediators promote angiogenesis and upregulate HIF-1α signaling, creating structurally immature, leaky vessels that facilitate the intravasation of tumor cells into the circulation. This angiogenic imbalance is especially prominent in smokers and is a known driver of tumor invasiveness. Moreover, COPD itself is now recognized not only as a localized respiratory disorder but as a systemic inflammatory syndrome, often associated with endothelial dysfunction, arterial stiffness, and accelerated atherosclerosis. Structural and functional alterations in the vascular tree (including reduced nitric oxide bioavailability, increased endothelin-1 expression, and pro-coagulant status) are frequently observed in patients with moderate to severe COPD. These changes predispose to vascular remodeling, aneurysm formation, and chronic hypoperfusion, and may facilitate a microenvironment permissive for tumor growth and dissemination [34]. Interestingly, vascular damage in COPD patients is not limited to atherosclerosis. Medial thinning, elastin degradation, and focal aneurysmal dilatations have been documented in postmortem analyses of patients with advanced COPD, particularly in smokers with concurrent pulmonary hypertension. These vascular abnormalities may alter the distribution of pulmonary and systemic blood flow, enhancing the retention and survival of CTCs and contributing to the peculiar metastatic patterns observed in lung cancer patients with significant smoking history [156]. In parallel, endothelial cells can undergo a related process known as endothelial to mesenchymal transition (endo-EMT), in which vascular integrity is lost and mesenchymal-like phenotypes emerge. In vasculopathy and COPD patients, chronic inflammation, shear stress, and oxidative damage are potent inducers of endo-EMT, which may both promote vascular remodeling and facilitate tumor cell extravasation at distant sites. CTC adhesion and extravasation are facilitated not only by mechanical and immunologic factors, but also by stromal plasticity; EMT and endo-EMT both remodel the extracellular matrix and generate a permissive environment for neoplastic dissemination. Moreover, in some instances, the reverse process, known as mesenchymal to epithelial transition (MET), which is required for the growth of metastatic lesions at distant sites, can occur early on CTCs and even on circulating cancer stem cells in the bloodstream [157,158], thus enhancing metastatic colonization and growth. The activation on EMT/MET processes also acts on migrating cancer cells [159] by affecting their mechanical memory [160], namely, cell geometrical and mechanical history, ultimately acting on their fate and invasive capacity [161,162,163,164]. It is conceivable that these shape changes in vivo impact neoplastic immunity, although dedicated experiments are still required.

These observations could potentially have implications for metastasis, therapeutic delivery, and immune modulation. This complex scenario calls for integrated therapeutic strategies capable of targeting both the neoplastic process and the underlying vascular-inflammatory substrate [165].

#### 4.1.1. Vascular Physiopathology

The arterial walls are multi-layered systems, with each layer—the innermost tunica intima, middle tunica media, and outermost tunica adventitia—helping to maintain arterial physiology. The intima consists of an endothelial cell (EC) monolayer, contains smooth muscle fibrocells and free cells, and is directly exposed to the hemodynamic environment; it is anchored to the basal membrane through a pool of elastic fibers defined as an internal elastic lamina. The EC layer regulates biomacromolecule permeability and acts as a hemodynamic mechanosensor transducing fluid force into biochemical signals. The tunica media is composed primarily of vascular smooth muscle cells (vSMCs), elastic fibers and laminae, collagen fibers, and proteoglycans, which contribute in molecule trafficking and in orchestrating both physiological and pathological vascular functions, such as remodeling after injuries. The VSMCs regulate vessel contraction/dilation, modulating blood pressure in response to flow, and also produce a significant portion of extracellular matrix (ECM) molecules, such as elastin, collagen, fibrillin, and proteoglycans, which maintain arterial biomechanical and contractile integrity. The vasa vasorum may also be missing in the tunica media of the largest arteries, and the supply of oxygen probably occurs much more effectively from the lumen than from them, thanks to the high partial pressure of oxygen. The adventitia is mainly composed of collagen and continues into the perivascular connective tissue. The external elastic lamina is between the media and adventitia, which is mainly composed of type I collagen fibrils, organized to form collagen fibers and fibroblasts, providing strength and support [166]. Notably, in the largest arteries in the media, elastic fibers are arranged concentrically and joined together by elastic connection bands. This allows the high distensibility and recoil capacity of the vascular wall, whereas the frame of fibrillar collagen assures the length–tension properties. The anatomic composition and mechanical performance vary near vascular bifurcations, where the adventitia increases its thickness. The aorta, the largest artery, is defined as an elastic or conducting vessel, based on its relatively high elastin content which allows it to maintain a constant blood pressure gradient throughout the cardiac cycle. This phenomenon is termed Windkessel’s effect [167], the physiological phenomenon that based on resistance and compliance variations at the level of the large elastic arteries, allows modification of the discontinuous flow of the cardiac output into a more continuous flow, transforming the kinetic energy of the blood coming from the left ventricle into elastic potential energy.

#### 4.1.2. Blood Flow Rheology

Blood circulation induces into vessels three different types of mechanic energy: (i) tangential shear stress (**τ**), which exerts direct flow on endothelial cells; (ii) perpendicular or radial shear stress (**γ**), which is defined as the radial thrust of blood pressure; (iii) stress related to parietal stretching, which is associated with parietal wall deformation. A fluid is said to be Newtonian if the tangential stress (**τ**) is directly proportional to deformation rate (shear stress, **γ**), according to the formula **τ** = μ**γ**, where μ is the dynamic viscosity, a thermophysical property of the fluid, which depends only on the temperature and not on the flow. A fluid that does not present this property is defined as non-Newtonian and is described by the same equation but replacing the viscosity with the apparent viscosity (μ_app_), **τ** = μ_app_ **γ** [168]. The apparent viscosity is not constant and is no longer a thermophysical property of the fluid, but depends on the flow properties. Blood is not a Newtonian fluid but it is a complex suspension in which cells are immersed into a colloidal solution, the plasma [169]. Non-Newtonian effects of blood flow are considered negligible when examining the motion of blood in large vessels, but they can become relevant in those areas of flow reversal and/or separation characterized by low shear rate values. The main differences from the Newtonian model for blood flow are mainly found at low speeds, in the distribution of WSS, in recirculation zones, and in secondary flows [170].

#### 4.1.3. Smoke-Associated Vascular Disease

It is well known that smoking is a potent risk factor for clinically relevant peripheral artery disease (PAD), with a significant dose–response association [171,172,173]. Arterial vasculature is subjected to morphological and functional alterations due to cigarette smoking. Pathologic lesions involve the arterial wall and blood cells as well. The effects of smoke on microcirculation encompass alterations in the endothelium, platelet aggregation and adhesiveness, the nervous system, and metabolic changes, which, overall, alter the flow and tissue perfusion [174]. Moreover, two major compounds of cigarette smoke are capable of determining vascular damage: nicotine acts preferably on large arteries, and carbon monoxide on small arteries, although both compounds damage the vascular system [175]. In animal models, smoke exposure induces the formation of lipid plaques by impacting total cholesterol content, ultimately leading to atherosclerosis [176,177]. The most relevant role in the generation of atherosclerotic lesions is played by nicotine [178]. In addition to altering lipid metabolisms, cigarette smoke extracts are known to induce oxidative stress and inflammation, which contribute to the atherosclerotic process through the dysfunction of macrophages, smooth cells, and endothelial elements, which are implicated in the development of PAD [175,179,180].

### 4.2. Circulating Tumor Cell Dynamics in Vasculopathy

Circulating tumor cells (CTCs) face a hostile intravascular environment characterized by hemodynamic stress, endothelial barriers, and immune surveillance [181] (Figure 1), yet the interplay between these biomechanical constraints and cancer cell survival has not been adequately dissected in lung cancer. Recent studies have begun to elucidate how mechanical factors such as fluid shear stress (FSS) influence CTC survival. FSS has been shown to trigger epithelial–mesenchymal transition (EMT) through JNK signaling, promoting resistance to apoptosis and enhancing metastatic fitness [182]. Moreover, shear stress can induce nuclear expansion via histone acetylation, a process that appears essential for CTC viability during circulation [183]. The mechanical resilience of CTCs, including their ability to deform through capillary-sized vessels and interact dynamically with blood components such as platelets, further contributes to their metastatic potential [184]. Platelets, in particular, play a protective role by shielding CTCs from immune recognition and facilitating vascular adhesion and extravasation [185]. These findings highlight the underappreciated contribution of mechanobiological forces in lung cancer metastasis and suggest novel avenues for therapeutic intervention. The process of metastatic dissemination involves a series of tightly regulated and highly selective steps, among which the intravascular survival and migration of CTCs represent a major bottleneck [186]. Once detached from the primary tumor mass, CTCs must enter the bloodstream, withstand shear stress from blood flow, evade immune detection, and, ultimately, extravasate at distant sites [187]. Although the molecular characteristics of CTCs have been extensively studied, the physical and mechanical challenges encountered during their journey through the circulation have received comparatively limited attention [188]. Blood flow exposes CTCs to varying degrees of shear stress depending on the vessel type and the local hemodynamic conditions. In large arteries, high shear forces can induce cell deformation, membrane disruption, and mechanical apoptosis, while in capillary beds or venous systems, lower shear may facilitate cellular arrest and adhesion [189,190]. Moreover, the shape, stiffness, and deformability of CTCs influence their ability to navigate narrow capillaries and avoid entrapment or destruction [191]. These biomechanical parameters are further complicated by the interaction of CTCs with circulating platelets, leukocytes, and endothelial cells, forming heterotypic aggregates that may either protect tumor cells or contribute to their clearance [192,193].

Importantly, in patients with chronic vascular conditions such as smoking-induced vasculopathy, the intravascular landscape may be profoundly altered. Endothelial dysfunction, vessel wall thickening, and microvascular rarefaction can all change the physical forces acting on CTCs and the mechanical properties of the vascular niche [180,194]. Such alterations may impair the regular laminar flow and induce turbulent patterns that influence CTC arrest, rolling, and extravasation [195]. Moreover, a stiffened and inflamed endothelium may facilitate abnormal adhesion molecule expression, allowing neoplastic clones to interact more readily with the vessel wall and initiate the metastatic cascade [196]. A particularly overlooked and clinically relevant scenario is the presence of arterial aneurysms, especially in major vessels such as the aorta, iliac, and femoral arteries, which are not infrequent in heavy smokers or patients with coexisting COPD [156,197]. These aneurysmal dilatations represent zones of altered wall compliance and disturbed flow dynamics, often characterized by low and oscillatory shear stress, vortex formation, and prolonged blood residence times [198]. Such hemodynamic profiles are known to promote thrombus formation and endothelial activation but may also enhance the mechanical trapping and adhesion of CTCs [199,200]. From a pathophysiological standpoint, the interface between intraluminal thrombi, endothelial dysfunction, and turbulent flow within aneurysms could create a permissive microenvironment for metastatic cell docking and transmigration [201]. An intraluminal thrombus may act as a scaffold where platelets and neutrophils release cytokines and extracellular traps (NETs), contributing to local immunosuppression and promoting CTC survival [202,203]. Furthermore, the transition zones between aneurysmal and normal-caliber segments may represent focal points of flow deceleration and mechanical impaction, further amplifying the risk of CTC arrest [204].

These observations are not only of theoretical interest but hold significant clinical implications, with valuable insights into the anatomic distribution and hemodynamic profiles of aneurysms in LC patients, potentially identifying anatomical “hot spots” for metastatic seeding [205]. In selected cases, vascular imaging data could be integrated with circulating tumor DNA (ctDNA) or CTC analysis to correlate vascular abnormalities with metastatic burden [206]. Moreover, in the future, prophylactic or therapeutic vascular interventions aimed at modulating aneurysmal flow (e.g., endograft reshaping) could become part of a multidisciplinary strategy to limit systemic dissemination in high-risk oncologic patients [165]. Recent advances in computational modeling and in vitro flow-based systems have begun to shed light on these biomechanical interactions. Microfluidic devices mimicking capillary networks and blood flow conditions have been used to quantify the deformability and adhesive behavior of tumor cells under dynamic shear conditions [207,208]. Such platforms may be instrumental in deciphering how altered hemodynamics in vasculopathic patients affects the efficiency of metastatic spread and could lead to the identification of novel mechanical vulnerabilities in circulating cancer cells [209]. In patients with atherosclerotic disease, vessel wall stiffness and plaque formation lead to disturbed flow profiles characterized by low shear stress zones and vortex formation [28]. These microenvironments favor CTC margination, rolling, and eventual adhesion to the endothelial surface. Similarly, arterial aneurysms, which are increasingly detected in lung cancer patients, especially smokers, create localized areas of low velocity and high recirculation, acting as potential mechanical traps for CTCs [149]. Preliminary computational models suggest that such aneurysmal niches could serve as “metastatic filters,” wherein tumor cells are temporarily retained, exposed to altered oxygen gradients and cytokine profiles, and eventually driven to extravasate under local inflammatory and mechanical cues [204]. These regions may also sustain platelet-rich microthrombi, further enhancing tumor–endothelial interactions [201]. Advanced computational fluid dynamics (CFD), integrated with patient-specific vascular geometries obtained via CT–angiography or MRI, allows precise simulation of hemodynamic patterns in both normal and diseased vessels. In silico studies have successfully modeled flow and shear profiles within thoracic aneurysms, CTC trajectory and deformation under pulsatile pressure waves, and tumor cell adhesion under pro-inflammatory endothelial activation [204]. Recent hybrid models also incorporate multi-physics elements, combining mechanical stress with biochemical gradients (e.g., oxygen tension and cytokine fields) to predict the likelihood of metastatic seeding in given vascular districts [208]. These platforms may help identify high-risk vascular phenotypes, especially in patients with combined COPD and systemic vasculopathy [209].

It should be added that, although going beyond the scope of this review, which is centered on smoke, vasculopathy can be induced by exposure to other toxins, among them, metal elements, which have been detected in atherosclerotic plaques [210], in particular, cadmium, an environmental pollutant, closely linked to the development of atherosclerosis and hypertension and systemic oxidative stress [211]. Interestingly, we and others, have reported the association between COPD, aneurysm rupture, and exposure to environmental particulate matter (PM), PM2.5 and PM10 [212,213,214,215].

### 4.3. Apolipoproteins

A growing body of evidence implicates apolipoproteins (particularly ApoA-I and ApoE) as key modulators of lung cancer metastasis through both vascular and immune pathways. ApoA-I, the principal protein of high-density lipoprotein (HDL), exerts potent antioxidative and anti-inflammatory effects while promoting cholesterol efflux from macrophages and endothelial cells. In murine models, ApoA-I mimetic peptides have been shown to reprogram tumor-associated macrophages toward an M1-like phenotype, reduce secretion of pro-metastatic cytokines (e.g., IL-6 and TNF-α), and, ultimately, could suppress pulmonary metastatic burden [216]. By contrast, ApoE appears to facilitate neoplastic dissemination: elevated ApoE expression in lung cancer cells enhances adhesion to LRP1-expressing endothelium, promotes transendothelial migration under disturbed flow, and increases metastatic colonization in distal organs [217].

Importantly, patients with COPD and smoking-related vasculopathy often exhibit dysregulated lipoprotein profiles, characterized by reduced ApoA-I/HDL levels and increased ApoE/LDL ratios [218]; this may exacerbate endothelial dysfunction and alter local shear stress patterns. Such lipid imbalances can impair endothelial barrier integrity, promote low-shear niches in aneurysmal segments, and synergize with the inflammatory milieu to create “metastatic traps” for circulating tumor cells. Taken together, these findings position apolipoproteins at the crossroads of lipid metabolism, vascular mechanics, and immune modulation, suggesting their potential as both biomarkers of metastatic risk and targets for novel interventions within the immune–vascular axis.

## 5. The Immune–Inflammatory Axis in Vasculopathy and Lung Cancer Dissemination

The chronic immune–inflammatory state observed in both smoking-induced vasculopathy and lung malignancy creates biologically fertile ground for both tumor invasiveness and distant dissemination (Table 2).

In this setting, the immune system, rather than acting as a barrier, may paradoxically facilitate metastatic spread through a series of dysfunctional responses involving both innate and adaptive components [219,220,221,222,223] (Table 3).

At lung level, among innate immune players, neutrophils have a prominent role. Chronic inflammation—as that sustained by COPD—leads to persistent neutrophil activation and degranulation, with release of reactive oxygen species (ROS), matrix metalloproteinases (MMPs), and the formation of neutrophil extracellular traps (NETs). In murine models, NET-rich microenvironments were associated with enhanced metastatic colonization of the lung and liver [202]. Monocyte/macrophage polarization is also critical. In the vasculopathic lung, macrophages are often turned toward an M2-like phenotype, promoting tissue remodeling, immunosuppression, and tumor angiogenesis. These macrophages can release VEGF, TGF-β, and IL-10, factors that not only support tumor growth but also modulate endothelial behavior and vascular permeability [193]. The adaptive immune system is not spared. Chronic smoking and inflammation drive T-cell exhaustion, reduce cytotoxic CD8+ T-cell activity, and promote the expansion of regulatory T cells (Tregs). This shift in immune balance may reduce immune surveillance against disseminated tumor cells and promote their survival and proliferation at secondary sites. NETs have been shown to enhance CTCs’ invasive potential, protect them from immune clearance, and even promote their extravasation and metastatic outgrowth. Moreover, altered dendritic cell function in vasculopathic tissues limits effective antigen presentation, further weakening antitumor immunity [224].

Once detached by the primary mass, CTCs enter the blood flow, which cooperates in metastatic promotion. Central to this phenomenon is the endothelial cell, which serves as both a target and a mediator of inflammatory injury. Activated endothelium in vasculopathic patients expresses increased levels of adhesion molecules such as ICAM-1, VCAM-1, and E-selectin, promoting leukocyte trafficking and, inadvertently, facilitating the adhesion and transmigration of circulating tumor cells (CTCs). Simultaneously, endothelial permeability is increased under inflammatory conditions, easing the paracellular passage of tumor cells into distant tissues [225,226,227]. At the molecular level, the crosstalk between inflammatory cytokines (IL-6, IL-1β, and TNF-α), chemokines (CXCL12 and CCL2), and proangiogenic signals (VEGF-A and Ang2) orchestrates a permissive metastatic niche in vasculopathic environments. This network sustains the pre-metastatic conditioning of distant organs, paving the way for efficient seeding of tumor clones [228]. A particularly relevant concept is the “inflammatory pre-metastatic niche”, wherein systemic inflammation and vascular pathology create specific molecular and cellular landscapes in organs distant from the primary tumor, which become more receptive to metastatic colonization. As discussed above, this niche is characterized by endothelial activation, extracellular matrix remodeling, stromal cell recruitment, and immune suppression, all of which are exacerbated in patients with underlying vasculopathy. In this scenario, the vasculature is no longer a passive conduit but becomes an active facilitator of metastatic spread, both as a dysfunctional barrier and as a pro-inflammatory signal amplifier [136].

### Immune Checkpoint Inhibitors, Bronchodilation, and Anti-Platelet Therapy

The advent of immune checkpoint inhibitors (ICIs) has transformed the therapeutic landscape of non-small-cell lung cancer (NSCLC), offering prolonged survival even in advanced disease stages [229]. However, the efficacy and safety profile of ICIs can be significantly influenced by the underlying cardio-pulmonary and vascular status of the patient, including co-administered medications like bronchodilators and antiplatelet agents. Indeed, ICIs act by restoring cytotoxic T-cell function, primarily via blockade of the PD-1/PD-L1 and CTLA-4 axes [9,37,160,230]. However, vascular inflammation can modulate the tumor microenvironment in ways that interfere with ICI efficacy. Platelets are emerging as crucial players in cancer biology and secrete growth factors such as PDGF and TGF-β that enhance tumor cell survival. In vasculopathic patients, low-dose aspirin or platelet receptor P2Y_12_ inhibitors are frequently prescribed, and their anti-metastatic potential is receiving growing attention [231]. Triple bronchodilation (typically combining a long-acting β_2_-agonist (LABA), a long-acting muscarinic antagonist (LAMA), and an inhaled corticosteroid (ICS)) is the cornerstone of COPD management in patients with severe airflow limitation. These drugs, beyond their airway-specific effects, can have systemic implications relevant to tumor–immune dynamics [8,33,232]. Taken together, this pharmacologic triad, immunotherapy, bronchodilation, and antiplatelet therapy represents a highly dynamic system, in which reciprocal interactions may profoundly shape lung cancer outcomes. However, some relevant issues need to be implemented and clarified (Table 4). Understanding these interdependencies is essential for designing personalized treatment strategies in patients with complex vascular and respiratory comorbidities.

## 6. Conclusions

A smoking habit is known be strictly related and interconnected with the arousal of both arteriopathy and cancer, and a large number of experimental reports point out the biologic basis, linking smoke-induced malignant transformation and peripheral arterial disease. Although smokers most often develop cancer and chronic arteriopathy, these conditions are usually evaluated as distinct nosologic entities. This review proposes that the intersection of key pathological axes—chronic epithelial and stromal remodeling (driven by smoke), vascular dysfunction and altered hemodynamics, and immune dysregulation within hypoxic niches—together may cooperate to facilitate metastatic dissemination in patients with COPD-associated lung cancer. While some mechanisms remain hypothetical, this framework offers novel hypotheses for future translational research. By narrowing our focus to biologically plausible COPD-related vascular alterations, such as local hypoxia, inflammation, and endothelial injury, we provide a more integrated and disease-relevant framework that supports the hypothesis of vascular permissiveness to metastasis in lung cancer patients. While substantial research has focused on the genetic, epigenetic, and immunologic drivers of LC-spreading progression, the role of mechanical forces governing metastatic dissemination remains largely unexplored. Understanding the biomechanical and molecular dynamics of LC progression in altered vascular territories offers several translational opportunities: (i) risk stratification: CFD models could be used to predict regions at higher metastatic risk based on vascular topology and flow conditions [165]; (ii) surgical planning: in patients with aneurysms or localized vascular remodeling, identifying pro-metastatic hemodynamic niches may guide endovascular interventions aimed not only at preventing rupture, but also at disrupting potential metastatic highways [205]; (iii) therapeutic targeting: modulation of flow dynamics via antithrombotic therapy, endothelial normalization agents, or even implantable flow regulators could theoretically reduce CTC retention and dissemination.

Ultimately, bridging vascular surgery, oncology, and computational biology may open novel avenues in the prevention of metastatic disease, particularly in patients with complex comorbid profiles.

## Figures and Tables

**Figure 1 cells-14-01225-f001:**
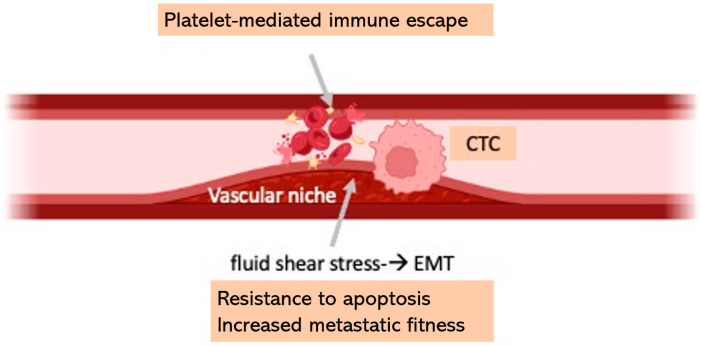
Selective advantages of CTCs in the vascular disease context. Created in BioRender.com.

**Table 1 cells-14-01225-t001:** Pro-metastatic mechanisms induced by smoke. Smoke affects early phases of LC onset in most cases in a COPD context. It is known that LC begins earlier and with multifocal lesions in Asian women and never-smokers. Adenocarcinoma has been the predominant oncotype in non-smokers, but currently its prevalence is also increasing in smokers (changes in the composition of cigarettes). The parenchymal site of growth of neoplastic lesions in smokers is both the peripheral and central lung; in non-smokers the sub-mantle areas are generally involved. Survival data (corrected for known prognostic factors) are more favorable for non-smokers [152,153,154,155]. uPA: urokinase plasminogen activator; OPN: osteopontin. ↑: increased; ↓: reduced.

Pro-Metastatic Smoke-Related Mechanisms	Refs.
↑ invasive g rowth program mediated by scatter factors	[105,106]
↑ uPA through activation of EGFR/ERK1/MAPK-mediated signaling	[107]
↑ OPN expression via the JAK2/STAT3 pathway; attraction of mesenchymal stromal cells	[108]
↓ NK cell function; altered innate immunity	[109]
Nicotine-mediated selective advantage and prevention of drug-mediated apoptosis	[110]
Chronic inflammation (↑ NF-κB, COX-2, 5-LOX)	[111,112]
Smoke-related IL-6-associated bone metastasization	[113]
Altered microbioma	[114]
Deregulation of EV and exosomes and the subsequent release of pro-metastatic RNAs	[115,116,117]

**Table 2 cells-14-01225-t002:** Point for clinical practice. Risk factors, mechanisms, and potential therapeutic targets of metastatic dissemination in smoke-induced pathologic blood vessels. TNF: tumor necrosis factor; CSF-1R: receptor of colony-stimulating factor 1; ROS: reactive oxygen species, NO: nitric oxide; IFP: interstitial fluid pressure; MMP: matrix metalloproteinase; ECM: extracellular matrix; WSS: wall shear stress; DNMT: DNA methyltransferase; HDAC: histone deacetylase.

l	Factor	Mechanism	Potential Actionable Target
**Immune–Inflammatory**	IL-1β, TNF-α, IL-6	Pro-inflammatory cytokines promoting chronic inflammation, tumor promotion, angiogenesis	Anti-cytokine therapies (e.g., IL-1β inhibitors, anti-TNF agents)
	NF-κB, STAT3, HIF-1α signaling	Sustained pro-survival and inflammatory signaling pathways	NF-κB/STAT3 inhibitors, HIF-1α modulators
	T-reg depletion, CD8+ T-cell predominance	Immune imbalance, reduced immunosurveillance	T-reg restoration, immune checkpoint modulation
	Macrophage polarization (M1 dominance, tumor-associated macrophages)	Pro-tumor inflammation, matrix remodeling	CSF-1R inhibitors, macrophage reprogramming
**Oxidative stress**	ROS, mitochondrial dysfunction	DNA damage, impaired apoptosis	Antioxidants, mitochondrial protective agents
**Vascular dysfunction**	Endothelial adhesion molecules (VCAM-1, ICAM-1, E-selectin)	Promotes leukocyte adhesion, CTC arrest	Anti-adhesion therapies (e.g., selectin blockers)
	Reduced NO bioavailability	Endothelial dysfunction, impaired vasodilation	NO donors, endothelial stabilizers
	Pathologic angiogenesis (VEGF, Angiopoietin-2)	Abnormal, leaky vessels facilitating metastasis	Anti-VEGF therapies, angiopoietin pathway inhibitors
**Biomechanical**	Low shear stress, turbulent flow, flow stagnation	Facilitates CTC adhesion, extravasation	Vascular normalization, flow modulation
	Elevated IFP	Drives outward migration of tumor cells	Anti-VEGF, normalization of tumor IFP
**Extracellular matrix**	MMP-2, MMP-9	ECM degradation enabling invasion	MMP inhibitors
**Epigenetic/Genetic**	DNA methylation, histone modifications	Silencing of tumor suppressor genes	Epigenetic drugs (e.g., DNMT inhibitors, HDAC inhibitors)
**Apolipoproteins/Lipids**	Oxidized LDL, ApoB/ApoE dysregulation	Endothelial activation, macrophage recruitment	Lipid-lowering agents, ApoE modulators
**Aneurysmal niche**	MMP overexpression, inflammatory cell infiltration	Vessel wall degradation, permissive microenvironment	MMP inhibitors, anti-inflammatory therapies
	Hemodynamic abnormalities in aneurysm (low WSS, recirculation zones)	Increased CTC residence time and adhesion	Flow-altering endovascular interventions
**Pharmacologic interactions**	Corticosteroids, ICSs	Immune suppression, impaired antigen presentation	Tapering strategies, ICS alternatives
	Aspirin, P2Y12 inhibitors	Reduce platelet cloaking of CTCs, inhibit thrombosis	Consider as adjunct to anti-metastatic therapy
	Triple inhaled therapy (ICS/LABA/LAMA)	Reduces inflammation, improves oxygenation	May indirectly mitigate hypoxia-driven tumor progression
**PD-L1 expression**	Immune evasion by inhibiting T cell-mediated cytotoxicity	Enhances tumor immune escape and metastatic spread	Anti-PD-1/PD-L1 therapy (e.g., pembrolizumab)
**Tumor–stromal crosstalk**	EMT promotion (TGF-β, matrix degradation)	Increases invasiveness, resistance to apoptosis	TGF-β inhibitors, EMT blockers

**Table 3 cells-14-01225-t003:** Relationship between smoking and tumor progression at both lung and vascular level. ↑: increased; ↓: reduced.

Smoke	Cells	Mechanism	Molecules/Mediators	Effects
**Lung**	Neutrophils	Activation–Degranulation	ROS, MMP, NET	↑ Invasive capacity
	Macrophages	M2 polarization	VEGF, TGF-β, IL-10	Tissue remodeling, immunosuppression, tumor angiogenesis
	T lymphocytes	↓ Cytotoxic CD8+ T ↑ Tregs		↓ Immune surveillance ↑ Invasive capacity
**Vessels**	Endothelial cells	Activation ↑ Permeability	ICAM-1, VCAM-1, E-selectin	↑Adhesion, transmigration of CTCs
**Vascular Niche**	Endothelial cells, pericytes, smooth muscle cells, inflammatory cells, stem cells	Systemic inflammation, vascular disease	Cytokine: IL-6, IL-1β, TNF-α Chemokines: CXCL12, CCL2 Angiogenic factors: VEGF-A, Ang2	↑ Paracellular passage of tumor cells to distant tissues

**Table 4 cells-14-01225-t004:** Known key points and gaps of knowledge regarding synergy and mutual interferences between immune checkpoint inhibitors, triple bronchodilators, and antiplatelet agents.

Key Points	Knowledge Gap
**LUNG CANCER → Immune checkpoint inhibitors** Inflamed or remodeled vasculature in COPD or systemic vasculopathy may be a critical determinant of response to immunotherapy.	
Vascular inflammation can modulate the TME in ways that interfere with ICI efficacy. Chronically inflamed vessels release VEGF, IL-6, and TGF-β, all of which are known to foster immune exclusion, impede T-cell infiltration, and promote an immunosuppressive milieu. Endothelial cells express immune regulatory molecules, such as PD-L1, FasL, and Galectin-9, which can engage immune checkpoints and inhibit T-cell trafficking	Lack of validation of predictive value of vascular biomarkers (e.g., circulating angiopoietins, soluble VCAM-1) in anticipating ICI responsiveness Lack of data regarding the role of ICS on systemic vasculopathy
**COPD → Triple bronchodilation** Beyond airway-specific effects, it can have systemic implications relevant to tumor–immune dynamics.	
LABA: β_2_-agonists may exert mild immunomodulatory effects by modulating cytokine release and T-cell activation. LAMA: Muscarinic antagonists, especially tiotropium, have shown potential anti-inflammatory properties in preclinical lung cancer models by reducing macrophage infiltration and oxidative stress. ICS: Inhaled corticosteroids, while effective in reducing exacerbations, may attenuate the pro-inflammatory signaling necessary for optimal ICI activity. Retrospective analyses suggest that high-dose ICS usage could correlate with lower ICI response rates, particularly in patients with low-tumor PD-L1 expression.	Clinical relevance of β_2_-agonists effects on T-cell activation and of LAMA effects remain controversial. Overall, the cumulative effect of triple bronchodilation on systemic immunity (specifically on the TME immune phenotype) warrants deeper investigation, especially in multimorbid LC patients undergoing ICI.
**VASCULOPATHY → Antiplatelet therapy** Platelets can shield CTCs from immune recognition and promote their extravasation.	
Reduction of metastatic burden, particularly when combined with ICIs, by decreasing platelet–tumor aggregates, inhibiting endothelial activation, reducing thrombo-inflammation and modulating the release of immunosuppressive factors Platelet-derived microparticles may interfere with ICI efficacy by altering immune cell recruitment and T-cell priming.	In selected patients, the addition of antiplatelet therapy could synergize with ICIs by restoring vascular–immune equilibrium and impairing CTC survival and transit. There is a lack of known variables to stratify patients.

## Data Availability

No new data were created or analyzed in this study. Data sharing is not applicable to this article.

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
