# Peer review of "From COPD to Smoke-Related Arteriopathy: The Mechanical and Immune–Inflammatory Landscape Underlying Lung Cancer Distant Spreading—A Narrative Review"

_cells, 2025, doi:10.3390/cells14161225_

Round 1

Reviewer 1 Report (Previous Reviewer 1)

Comments and Suggestions for Authors

The revised manuscript is well written.

Author Response

We really thank the Reviewer for careful revision and comments of our work which is now enriched in scientific quality

Reviewer 2 Report (Previous Reviewer 2)

Comments and Suggestions for Authors
  1. The authors have done quite a lot of work very quickly on this manuscript, enter taken into account quite a lot of the suggestions that were made. The part on COPD and lung cancer and now much more balanced and in accord with the literature. My main problem is still with two things:

The tendency to write very long, detailed and somewhat incoherent paragraphs which do not really follow any particular narrative, and

The highly speculative monologue on the possibilities of vasculopathy frequently associated with smokers and therefore COPD being important in metastatic LC. If the rationale all of this hypothesis, and it It should be emphasized that is only a hypothesis, it would follw that we should be seeing lots of metastases directly related to large vessels but this is disappearingly rare, if indeed it happens at all. On the other hand, the idea that small vessel changes within tissues might enhance the chance of metastases is an interesting one, but again a hypothesis that could be followed up.

  1. Back to COPD for a moment, there is good data that neo vascularization of epithelium as a part of EMT, the so-called type 3, is important in metastatic disease is not new. There are papers from the Sohal, Soltani, Mahmood group the authors should read on this.

  1. The authors are now dealing with EMT, and endo-MT quite well but they have missed out the very important step of mesenchymal to epithelial transition (MET) as being very important to the interaction between cancer cells and the movement of these between the circulation and peripheral tissues. There's quite a literature and good reviews on this and it's important in oncology.

  1. Once again, I think the authors are overdoing the idea that micro tissue hypoxia is important even when the whole body is well oxygenated. I will drop all this stuff around HIFs, again to limit the amount of speculation that they're indulging in. They should not be using the extreme of end stage COPD to generalize to the average a person and the vast majority who develops LC.

  1. There are still a number of typos, but again I think the rewriting process needs to take much more time and be a serious attempt to cut out the enormous amount of speculation and to continue writing a solid piece of work. The current version is promising that this can be achieved.

Author Response

The authors have done quite a lot of work very quickly on this manuscript, enter taken into account quite a lot of the suggestions that were made. The part on COPD and lung cancer and now much more balanced and in accord with the literature.

We really thank the Reviewer for this critical and careful support to our work.

The tendency to write very long, detailed and somewhat incoherent paragraphs which do not really follow any particular narrative, and

The highly speculative monologue on the possibilities of vasculopathy frequently associated with smokers and therefore COPD being important in metastatic LC. If the rationale all of this hypothesis, and it It should be emphasized that is only a hypothesis, it would follw that we should be seeing lots of metastases directly related to large vessels but this is disappearingly rare, if indeed it happens at all. On the other hand, the idea that small vessel changes within tissues might enhance the chance of metastases is an interesting one, but again a hypothesis that could be followed up.

We thank the Reviewer for pointing out this issue. The hypotethical approach has been stressed in line 105 and line 132. However, it should be underlined that the experimental published data available till now do not allow the conclusion that the vessel dimensions are directly associated to metastatic frequency but, more properly, that pathologic vascular diameter and biologic alterations ( as in smoke-related chronic conditions) impact on blood rheology and biology and ultimately on the fate of circulating tumor cells.

Back to COPD for a moment, there is good data that neo vascularization of epithelium as a part of EMT, the so-called type 3, is important in metastatic disease is not new. There are papers from the Sohal, Soltani, Mahmood group the authors should read on this.

We thank the REviewer for this comment and the text has been implemented as follows: Thus, EMT can occur in physiological conditions, as during embryo development, namely the type I EMT [] or during wound healing, tissue regeneration and organ fibrosis (type II EMT) []. Aberrant activation of EMT (type-3 EMT) [] orchestrates cancer progression in terms of promotion of clonal evolution but also in generation a pro-metastatic stroma. Smoke is implicated in EMT activation in both lung cancer and COPD []. In this context, inhaled corticosteroids (ICS), mainly at higher doses, should exters a mitigation on LC risk reduction although in vivo studies are needed [].

The authors are now dealing with EMT, and endo-MT quite well but they have missed out the very important step of mesenchymal to epithelial transition (MET) as being very important to the interaction between cancer cells and the movement of these between the circulation and peripheral tissues. There's quite a literature and good reviews on this and it's important in oncology.

We agree with this comment and have modified the text as follows: Moreover, in some instances, the reverse process known as mesenchymal-to-epithelial transition (MET) which is required to the growth of metastatic lesions in distant sites, can occur early on CTC and even on circulating cancer stem cells in the bloodstream [] thus enhancing metastatic colonization and growth. The activation on EMT/MET processes also acts on migrating cancer cells [] by affecting their mechanical memory [], namely cell geometrical and mechanical history, ultimately acting on their fate and invasive capacity []. It is conceivable that these shape changes in vivo impact on neoplastic immunity, although dedicated experiments are still required. 

Once again, I think the authors are overdoing the idea that micro tissue hypoxia is important even when the whole body is well oxygenated. I will drop all this stuff around HIFs, again to limit the amount of speculation that they're indulging in. They should not be using the extreme of end stage COPD to generalize to the average a person and the vast majority who develops LC.

We thank the Reviewer for this comment and the section regarding hypoxia has been mostly reduced.

There are still a number of typos, but again I think the rewriting process needs to take much more time and be a serious attempt to cut out the enormous amount of speculation and to continue writing a solid piece of work. The current version is promising that this can be achieved.

We revised the text regarding typo errors and also coherently with other Reviewers' suggestions

[i] Kalluri R, Weinberg RA. The basics of epithelial-mesenchymal transition. J Clin Invest. 2009 Jun;119(6):1420-8. doi: 10.1172/JCI39104. Erratum in: J Clin Invest. 2010 May 3;120(5):1786.

[ii] Marconi GD, Fonticoli L, Rajan TS, Pierdomenico SD, Trubiani O, Pizzicannella J, Diomede F. Epithelial-Mesenchymal Transition (EMT): The Type-2 EMT in Wound Healing, Tissue Regeneration and Organ Fibrosis. Cells. 2021 Jun 23;10(7):1587. doi: 10.3390/cells10071587.

[iii] Mahmood M.Q., Sohal S.S., Shukla S.D. Epithelial mesenchymal transition in smokers: large versus small airways and relation to airflow obstruction. Int. J. Chron. Obstruct Pulmon. Dis. 2015;10:1515–1524. doi: 10.2147/COPD.S81032

[iv] Sohal SS. Chronic Obstructive Pulmonary Disease (COPD) and Lung Cancer: Epithelial Mesenchymal Transition (EMT), the Missing Link? EBioMedicine. 2015 Oct 17;2(11):1578-9. doi: 10.1016/j.ebiom.2015.10.016.

[v] Soltani A, Mahmood MQ, Reid DW, Walters EH. Cancer-protective effects of inhaled corticosteroids in COPD are likely related to modification of epithelial activation. Eur Respir J. 2019 Sep 12;54(3):1901088. doi: 10.1183/13993003.01088-2019

[vi] Hamilton G, Rath B. Mesenchymal-Epithelial Transition and Circulating Tumor Cells in Small Cell Lung Cancer. Adv Exp Med Biol. 2017;994:229-245. doi: 10.1007/978-3-319-55947-6_12.

[vii] Pei D, Shu X, Gassama-Diagne A, Thiery JP. Mesenchymal-epithelial transition in development and reprogramming. Nat Cell Biol. 2019 Jan;21(1):44-53. doi: 10.1038/s41556-018-0195-z.

[viii] Gostomczyk K, Marsool MDM, Tayyab H, Pandey A, Borowczak J, Macome F, Chacon J, Dave T, Maniewski M, Szylberg Ł. Targeting circulating tumor cells to prevent metastases. Hum Cell. 2024 Jan;37(1):101-120. doi: 10.1007/s13577-023-00992-6. 

[ix] Cambria E, Coughlin MF, Floryan MA, Offeddu GS, Shelton SE, Kamm RD. Linking cell mechanical memory and cancer metastasis. Nat Rev Cancer. 2024 Mar;24(3):216-228. doi: 10.1038/s41568-023-00656-5.

[x] Malinverno C, Corallino S, Giavazzi F, Bergert M, Li Q, Leoni M, Disanza A, Frittoli E, Oldani A, Martini E, Lendenmann T, Deflorian G, Beznoussenko GV, Poulikakos D, Haur OK, Uroz M, Trepat X, Parazzoli D, Maiuri P, Yu W, Ferrari A, Cerbino R, Scita G. Endocytic reawakening of motility in jammed epithelia. Nat Mater. 2017 May;16(5):587-596. doi: 10.1038/nmat4848.

[xi] Yang C, Tibbitt MW, Basta L, Anseth KS. Mechanical memory and dosing influence stem cell fate. Nat Mater. 2014 Jun;13(6):645-52. doi: 10.1038/nmat3889.

[xii] Li Y, Tang W, Guo M. The Cell as Matter: Connecting Molecular Biology to Cellular Functions. Matter. 2021 Jun 2;4(6):1863-1891. doi: 10.1016/j.matt.2021.03.013.

[xiii] D'Aniello C, Cermola F, Palamidessi A, Wanderlingh LG, Gagliardi M, Migliaccio A, Varrone F, Casalino L, Matarazzo MR, De Cesare D, Scita G, Patriarca EJ, Minchiotti G. Collagen Prolyl Hydroxylation-Dependent Metabolic Perturbation Governs Epigenetic Remodeling and Mesenchymal Transition in Pluripotent and Cancer Cells. Cancer Res. 2019 Jul 1;79(13):3235-3250. doi: 10.1158/0008-5472.CAN-18-2070.

Reviewer 3 Report (New Reviewer)

Comments and Suggestions for Authors

The review is nicely written and topic is of broad interest.

Consider adding a pargraph regarding increased cadmium load and role of Cd in the vasculophaty, which happens in COPD patient and smokers. COPD persons have several times higher Cd level relative to non-COPD individuals.

Author Response

We thank the Reviewer for pointing out this comment and the text has been implemented as follows: 

It should be added that, although going beyond the scope of this review which is centered on smoke, vasculopathy can be induced by the exposure of other toxics. Among them metal elements which have been detected in atherosclerotic plaques [[i]]. In detail cadmium, an environmental pollutant, closely linked with the development of atherosclerosis and hypertension and systemic oxidative stress [[ii]]. Interestingly, we and others, have reported the association between COPD, aneurysms rupture and exposure to environmental particulate matter (PM) PM2.5 and PM10 [[iii],[iv],[v],[vi]].

[i] Huang L, Liu Y, Yu L, Cheng A, Cao J, Wang R, Liu Y, Song S, Zhao W, Liu Q, Zhang D. Association of metal elements deposition with symptomatic carotid artery stenosis and their spatial distribution in atherosclerosis plaques. Metallomics. 2025 Jul 9;17(7):mfaf019. doi: 10.1093/mtomcs/mfaf019.

[ii] Angeli JK, Cruz Pereira CA, de Oliveira Faria T, Stefanon I, Padilha AS, Vassallo DV. Cadmium exposure induces vascular injury due to endothelial oxidative stress: the role of local angiotensin II and COX-2. Free Radic Biol Med. 2013 Dec;65:838-848. doi: 10.1016/j.freeradbiomed.2013.08.167

[iii] Bozzani A, Arici V, Cutti S, DI Marzo L, Sterpetti AV. Increased rupture of Abdominal Aortic Aneurysm in patients with COPD correlates with high atmospheric levels of PM2.5 and PM10. Int J Cardiol Cardiovasc Risk Prev. 2024 Mar 21;21:200266. doi: 10.1016/j.ijcrp.2024.200266.

[iv] Bozzani A, Cutti S, Marzo LD, Gabriele R, Sterpetti AV. Spatio-temporal correlation between admissions for ruptured abdominal aortic aneurysms and levels of atmospheric pollution in Italy. Curr Probl Cardiol. 2024 Feb;49(2):102249. doi: 10.1016/j.cpcardiol.2023.102249.

[v] Zha H, Wang R, Feng X, An C, Qian J. Spatial characteristics of the PM2.5/PM10 ratio and its indicative significance regarding air pollution in Hebei Province, China. Environ Monit Assess. 2021 Jul 10;193(8):486. doi: 10.1007/s10661-021-09258-w.

[vi] Urbanek T, Juśko M, Niewiem A, Kuczmik W, Ziaja D, Ziaja K. The influence of atmospheric pressure on aortic aneurysm rupture--is the diameter of the aneurysm important? Kardiol Pol. 2015;73(12):1327-33. doi: 10.5603/KP.a2015.0092

Round 2

Reviewer 2 Report (Previous Reviewer 2)

Comments and Suggestions for Authors

The manuscript is certainly much improved even if still over long. If the authors could restrict their discussion in vascular disease essentially around atheroma in smoking and COPD and downplay the hypothetical potential of vasculopathy being important in metastases.

By all means say that vasculopathy is something that could be researched in the context of tissue metastases, but how that would fit with the commonplaces that LC metastases go to, especially liver and brain I have no idea. That just seems to be far too much text something that still seems pretty unlikely.

The authors seemed keen to sustain a focus on “hypoxia”, but they could at least make the point that systemic hypoxemia is not a feature in COPD accept in advanced cases. In smokers, raised carbon monoxide levels could conceivably cause some tissue hypoxia because of reduced oxygen carriage in the blood in spite of good saturations, but again data would be required to back this up. There continues to be just too much space on speculation.

On line 577, "toxics" should be "toxins"

This manuscript is a resubmission of an earlier submission. The following is a list of the peer review reports and author responses from that submission.

Round 1

Reviewer 1 Report

Comments and Suggestions for Authors

Thank you for the opportunity to review your manuscript. I found your study on the association between smoking and lung cancer both insightful and highly relevant, and I read it with great interest. To further improve the clarity and impact of your work, I would like to offer a few suggestions:

  1. Clarity and Structure of Chapters

While each chapter is rich in detail, the density of information at times makes it difficult for readers to track the core arguments. I recommend adding a brief summary (小括) or incorporating a diagram/table at the end of each chapter to synthesize key findings. This would help readers consolidate complex content and facilitate smoother transitions between sections. For example, in chapters dealing with molecular mechanisms or intracellular signaling pathways, a schematic diagram could effectively visualize these relationships, making the content more accessible and intuitive.

  1. Logical Flow in the Conclusion: Smoking, Lung Cancer, and Vascular Involvement

The conclusion's discussion on the relationship between smoking, lung cancer, and vascular pathology would benefit from a clearer logical flow. At present, the central message regarding how vascular involvement mediates the effects of smoking on lung cancer is somewhat diffuse. I recommend that the manuscript more explicitly delineate how smoking-induced vascular changes contribute mechanistically to carcinogenesis and tumor progression.

To support this, consider including a conceptual figure summarizing:

  • The impact of smoking on vascular structures,
  • The intermediary mechanisms linking vascular dysfunction to tumor biology, and
  • The implications for lung cancer development and progression.

This visual summary could help clarify the authors’ central thesis and strengthen the overall coherence of the manuscript.

Author Response

Thank you for the opportunity to review your manuscript. I found your study on the association between smoking and lung cancer both insightful and highly relevant, and I read it with great interest.

We really thank the Reviewer for careful revision of the manuscript and for fruitful suggestion and comments to improve its scientific quality.

To further improve the clarity and impact of your work, I would like to offer a few suggestions:

  1. Clarity and Structure of Chapters

While each chapter is rich in detail, the density of information at times makes it difficult for readers to track the core arguments. I recommend adding a brief summary (小括) or incorporating a diagram/table at the end of each chapter to synthesize key findings. This would help readers consolidate complex content and facilitate smoother transitions between sections. For example, in chapters dealing with molecular mechanisms or intracellular signaling pathways, a schematic diagram could effectively visualize these relationships, making the content more accessible and intuitive.

We thank re Reviewer for this sugegstion and the text has been modified coherently. Paragraphs have been subdivided and Fig.2 has been added at sec. 4.4.

  1. Logical Flow in the Conclusion: Smoking, Lung Cancer, and Vascular Involvement

The conclusion's discussion on the relationship between smoking, lung cancer, and vascular pathology would benefit from a clearer logical flow. At present, the central message regarding how vascular involvement mediates the effects of smoking on lung cancer is somewhat diffuse. I recommend that the manuscript more explicitly delineate how smoking-induced vascular changes contribute mechanistically to carcinogenesis and tumor progression.

To support this, consider including a conceptual figure summarizing:

  • The impact of smoking on vascular structures,
  • The intermediary mechanisms linking vascular dysfunction to tumor biology, and
  • The implications for lung cancer development and progression.
  • This visual summary could help clarify the authors’ central thesis and strengthen the overall coherence of the manuscript.

We agree with this comment . The conclusion has been restructured and Tab.2 has been added to summarize the most relevant effects of smoke on lung and vasculature.

Reviewer 2 Report

Comments and Suggestions for Authors

  1. I enjoyed reading this paper; however, I don't think that a review on the relationship between COPD and lung cancer and its strong tendency to metastasize early on the one hand, and the long piece on vascular anatomy and physiology on the other hand, works at all well together. Further, the attempt to link metastases with atheromatous vascular disease and/or aneurysms for that matter is highly speculative at best, and well off the central topic.

  1. The fact that it is COPD, or indeed fixed airflow obstruction short of a clinical diagnosis, which is especially related to lung cancer and also to cardiac events is not really spelt out, except perhaps oddly in a brief phrase in the abstract. There is no adequate discussion on why such an important relationship or set of relationships exist, including for example physical constraints due to large pressure fluctuations on the heart and large vessels.
  1. A lot of separate ideas are packed into each long paragraph. It would be really nice to see this work set out with each individual idea put into a different paragraph or ideally under separate subheadings, and then worked up logically and readably. Definitions, abbreviations and various terms used need to be much more comprehensive.
  1. The review is dominated and almost obsessed by the overarching concept of “inflammation”, which is over-stretched in a definitional sense, without any pathological coherence. This may be somewhat philosophical, but I would stick with inflammation being an activation of the immune system, be it innate and/or adaptive. Other cells and tissues can respond to a variety of stimuli and can interact mutually with inflammation, such as epithelial and endothelial layers but that is not inflammation in itself, although may well be part of an integrated organ, tissue or systemic response.
  1. In COPD, there is certainly innate immune activation with lots of macrophages and some polymorphs in the airway lumen in particular, (but not so much in the sub epithelial airway wall), and there might be some activation of the adaptive system especially late in the process, but there is also an important role for microbial colonisation/infection from the earliest phases. Further, airway remodelling is a major and central feature even in early disease, with transformation of the epithelium itself but also development of EMT in basal stem cells related to subsequent myofibroblast proliferation and laying down about normal sub epithelial matrix proteins; there is thickening of all layers constituting the sub-epithelial structures. EMT is given several brief but not connected mentions in this review article, but it is never defined or described in any sort of detail, nor is its importance in COPD nor in the pathogenesis of lung cancer and the metastatic behaviour of lung cancer cells. The whole interacting EMT/endo-MetMT Story is completely missing.
  1. The genetics and epigenetics sections are interesting but again very poorly put together or summarised. The gene reprogramming of airway basal stem cells is certainly vital in COPD epithelial and sub epithelial pathology, and is very likely to be under epi-genetic control, but this needs to be teased out and described systematically and in a readable, coherent fashion.

  1. Hypoxia is brought into the story at one point around line 420, and yet apart from very severe COPD, hypoxemia is not a feature of the disease. So, the authors need to define what they mean by this concept in their storyline. It is yet another rather orphan section, interesting in itself, but highly disjointed and not necessarily terribly accurate. Different topics keep appearing and disappearing almost at random. This is rather reflected in the final conclusions, which are vague to say the least and perhaps reflect the fact that there is no coherent narrative actually developed in the article.

  1. The English in the writing is pretty good. There are a few typos and small grammatical errors, such as getting the singular versus plural verb mixed up, and on line 290 instead of “alter” there should be the gerund “altering”. Hopefully some of this will come out in this substantial rewriting exercise did I suggest now occurs.

Author Response

We thank the Reviewer for careful revision of our text and for the fruitful comments and suggestions. As suggested the structure of the review has been strongly revised. 

Comment 1. The fact that it is COPD, or indeed fixed airflow obstruction short of a clinical diagnosis, which is especially related to lung cancer and also to cardiac events is not really spelt out, except perhaps oddly in a brief phrase in the abstract. There is no adequate discussion on why such an important relationship or set of relationships exist, including for example physical constraints due to large pressure fluctuations on the heart and large vessels.

Answer 1 to Comment 1. We agree with this comment and a dedicated paragraph has beed added ( 3.1.3)

Comment 2. A lot of separate ideas are packed into each long paragraph. It would be really nice to see this work set out with each individual idea put into a different paragraph or ideally under separate subheadings, and then worked up logically and readably. Definitions, abbreviations and various terms used need to be much more comprehensive.

Answer 2 to Comment 2. WE thank the Reviewer for this suggestion. The structure of the text has been implemented with sub-headings , abbreviations and a figure and a table ( also coherently with Rev.1 comments )

3. Comment 3. The review is dominated and almost obsessed by the overarching concept of “inflammation”, which is over-stretched in a definitional sense, without any pathological coherence. This may be somewhat philosophical, but I would stick with inflammation being an activation of the immune system, be it innate and/or adaptive. Other cells and tissues can respond to a variety of stimuli and can interact mutually with inflammation, such as epithelial and endothelial layers but that is not inflammation in itself, although may well be part of an integrated organ, tissue or systemic response

Answer 3 to comment 3. Although inflammation is a key issue in the review, we thank the Reviewer for pointing out this criticism. The text has been implement in section 3.1.1

Comment 4. In COPD, there is certainly innate immune activation with lots of macrophages and some polymorphs in the airway lumen in particular, (but not so much in the sub epithelial airway wall), and there might be some activation of the adaptive system especially late in the process, but there is also an important role for microbial colonisation/infection from the earliest phases. Further, airway remodelling is a major and central feature even in early disease, with transformation of the epithelium itself but also development of EMT in basal stem cells related to subsequent myofibroblast proliferation and laying down about normal sub epithelial matrix proteins; there is thickening of all layers constituting the sub-epithelial structures. EMT is given several brief but not connected mentions in this review article, but it is never defined or described in any sort of detail, nor is its importance in COPD nor in the pathogenesis of lung cancer and the metastatic behaviour of lung cancer cells. The whole interacting EMT/endo-MetMT Story is completely missing.

Answer 4 to Comment 4. We really thank the Reviewer for pointing out this critical issue. The text has been implemented iaccording to Reviewer's commebtn section 3.1.1 and 4.1 

Comment 5. The genetics and epigenetics sections are interesting but again very poorly put together or summarised. The gene reprogramming of airway basal stem cells is certainly vital in COPD epithelial and sub epithelial pathology, and is very likely to be under epi-genetic control, but this needs to be teased out and described systematically and in a readable, coherent fashion. 

Answer 6 to Comment 6. We agree with this suggestion and section 3.1.2 has been added. 

Comment 7. Hypoxia is brought into the story at one point around line 420, and yet apart from very severe COPD, hypoxemia is not a feature of the disease. So, the authors need to define what they mean by this concept in their storyline. It is yet another rather orphan section, interesting in itself, but highly disjointed and not necessarily terribly accurate. Different topics keep appearing and disappearing almost at random. This is rather reflected in the final conclusions, which are vague to say the least and perhaps reflect the fact that there is no coherent narrative actually developed in the article.

Answer 7 to Comment 7. We agree with this comment and the role oh hypoxia has been better clarified in through the whole text. 

Comment 8. The English in the writing is pretty good. There are a few typos and small grammatical errors, such as getting the singular versus plural verb mixed up, and on line 290 instead of “alter” there should be the gerund “altering”. Hopefully some of this will come out in this substantial rewriting exercise did I suggest now occurs

Answwer 8 to comment 8. We thank the Reviewer for careful reading and the text has been revised.